# DYNASEAL: A BACKEND-CONTROLLED LLM API KEY DISTRIBUTION SCHEME WITH CONSTRAINED INVOCATION PARAMETERS

**Jiahao Zhao[1], Jiayi Nan[3], Lai Wei[4], Yichen Yang[5]**
Xi'an University of Posts and Telecommunications
`{zjh, nanjy0108, 1606270965, 2647797263}@stu.xupt.edu.cn`

**Fan Wu[2]**
Xi'an Jiaotong University
`wfgods@gmail.com`

## ABSTRACT

The proliferation of edge-device interactions with cloud-based Large Language Models (LLMs) has exposed critical security vulnerabilities in traditional authentication methods like static Bearer Tokens. Existing solutions—pre-embedded API keys and server relays—suffer from security risks, latency, and bandwidth inefficiencies. We present **Dynaseal**, a secure and efficient framework that empowers backend servers to enforce fine-grained control over edge-device model invocations. By integrating cryptographically signed, short-lived JWT tokens with embedded invocation parameters (e.g., model selection, token limits), Dynaseal ensures tamper-proof authentication while eliminating the need for resource-heavy server relays. Our experiments demonstrate up to 99% reduction in backend traffic compared to relay-based approaches, with zero additional latency for edge devices. The protocol's self-contained tokens and parameterized constraints enable secure, decentralized model access at scale, addressing critical gaps in edge-AI security without compromising usability.

## 1 INTRODUCTION

Large Language Models (LLMs) (Hoffmann et al., 2022; Kaplan et al., 2020), such as ChatGPT (OpenAI, 2023), GPT-4 (OpenAI et al., 2023), and Claude 3 (Anthropic, 2024), have shown remarkable progress and impact across diverse domains (Brown et al., 2020). Current LLM API access relies on Bearer Token authentication, but this faces challenges with growing edge device inference needs. Edge devices include smartphones, PCs, and microcontrollers interfacing with cloud models.

Figure 1 illustrates the existing model invocation approaches. Two common approaches for edge device model invocation are:

- **Pre-embedded API Keys**(Shown in Figure 1a): API keys are configured in devices, enabling direct model access.
- **Server Relay**(Shown in Figure 1b): Intermediary servers relay requests, requiring persistent device-server connections.

Both approaches have limitations: pre-embedded API keys are vulnerable to security breaches, and server relay introduces latency and bandwidth overhead.

We propose **Dynaseal** (**Dyna**mic **Seal**), a solution that separates the business backend from large language model deployment. Through dynamically distributed API controls, it achieves an optimal trade-off between pre-embedded API keys and server relay approaches, making it suitable for practical implementation.

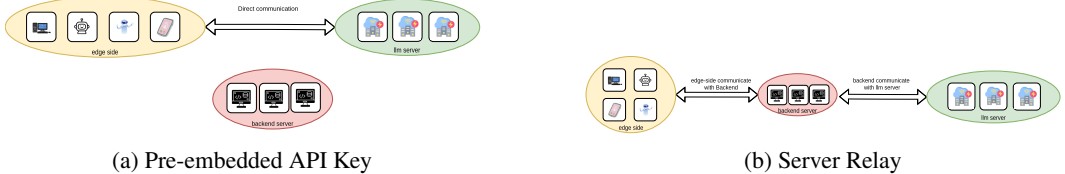

(a) Pre-embedded API Key                    (b) Server Relay

Figure 1: Current Model Invocation Methods

## 2 RELATED WORK

### 2.1 JWT TOKEN AND BEARER TOKEN AUTHENTICATION

JSON Web Token (JWT) represents a compact, URL-safe means of representing claims between parties (Jones et al., 2015b). A JWT consists of three parts: a header specifying the signing algorithm (Jones et al., 2015a), a payload containing claims, and a signature for verification. The self-contained nature of JWTs eliminates the need for database lookups, making them particularly efficient for stateless authentication. However, this approach also presents challenges in token revocation and session management, requiring additional mechanisms such as blacklisting or short expiration times.

Bearer token authentication is a widely adopted protocol for securing web APIs and services (Jones & Hardt, 2012). This mechanism allows clients to access protected resources by presenting a token, which serves as proof of authorization. The token is typically transmitted in the HTTP Authorization header with the "Bearer" scheme.

### 2.2 API KEY CONTROL

Some attempts have been made by the community to address this issue, but each has its limitations as shown in Table 2. The OpenAI API (OpenAI, 2024) does not provide server-side keys and can only use Bearer Tokens on the client side. Zhipu AI's keys (Zhipu, 2024) offer both server-side and client-side invocation methods, supporting server-issued keys and expiration control, but they cannot restrict critical parameters, leaving them vulnerable to attacks. Although OneAPI (songquanpeng, 2024) can redistribute keys, the invocation method remains Bearer Token-based, failing to resolve the client-side invocation problem.

## 3 METHOD

The system architecture comprises three primary components: Large Language Model (LLM) service providers, backend servers, and edge devices.

- The LLM service providers are organizations that host and operate large language models, being responsible for all model-generated responses and inference operations.

- The backend server is responsible for authenticating edge devices and handling business logic, with specific implementations determined by engineering scenarios. The backend server actually has a dual identity: on one hand, it is a user of the large model service provider and needs to authenticate itself to them. On the other hand, it acts as a service provider for edge devices and needs to authenticate these devices.

- Edge devices encompass a diverse range of hardware platforms, from sophisticated devices such as smartphones and personal computers to resource-constrained systems like microcontrollers.

### 3.1 BACKEND AUTHENTICATION

Prior to backend server deployment, a kv-pair (comprising user-id and secret-key) must be obtained from the LLM provider. The kv-pair are subsequently integrated into the service configuration for token generation and identity verification purposes.

The user-id is the backend's identity identifier with the large model service provider, while the secret-key is used to generate token signatures. It is safe for the user-id to be public as it is merely an identifier, whereas the secret-key remains confidential and is used for generating token signatures.

## 3.2 TOKEN STRUCTURE

The Dynaseal token consists of two parts: a parameter calling dictionary and a signature. The parameter calling dictionary contains key parameters for model invocation, such as model name and maximum token count, and includes a user-id field. The signature is generated using a secret-key pair, ensuring the token's integrity and preventing token tampering.

We adopted the widely used JWT token (Jones et al., 2015b) in network authentication and customized each part of the JWT TOKEN. The content is shown in Figure 2.

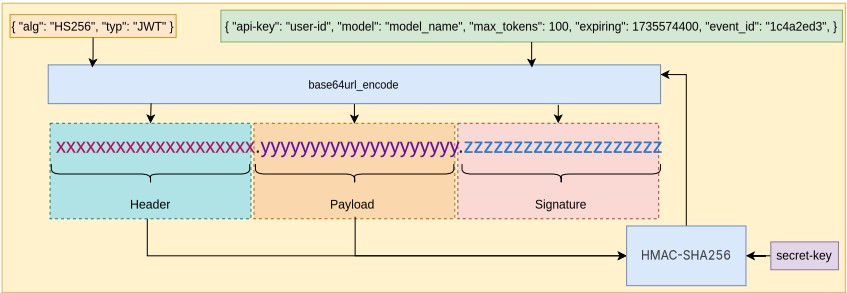

Figure 2: Dynaseal token Structure

The following explains each part of the JWT TOKEN:

- **Header**: Declare the encryption algorithm and token type.
- **Payload**: Include key parameters such as model name and maximum token count. The api-key field in payload is configured as user-id of the key-value pair to identify the backend server's identity with the large model provider. The expiration time is set extremely short (e.g., 1s) to prevent reuse.
- **Signature**: Sign with a key-value pair secret-key to ensure token integrity, preventing token tampering.

It should be noted that while we can only minimize the token validity period, we cannot completely eliminate the risk of token replay attacks. Therefore, there are two solutions:

- The large model service provider can record tokens and take appropriate action when token replay is detected, such as rejecting the request.
- Set an extremely short validity period. Although this may result in numerous timeouts under extreme network conditions, it can effectively prevent losses.

## 3.3 INTERACTION PROCESS

As shown in the 3, the backend server issues tokens to edge devices, with each token encapsulating critical model invocation parameters. Edge devices leverage these tokens to initiate model calls, while the LLM service infrastructure enforces strict invocation constraints based on the parameters embedded within the tokens. Upon completion of the model response, the backend system receives relevant notifications through established callback mechanisms, facilitating comprehensive request lifecycle management.

1. **Request for token**: Edge-side devices requests token from backend for subsequent interactions accorfind to business logic.
2. **Issue token**: Backend issues token to edge-side.
3. **Request response**: Edge-side uses token to request response insead of Bearer Token.

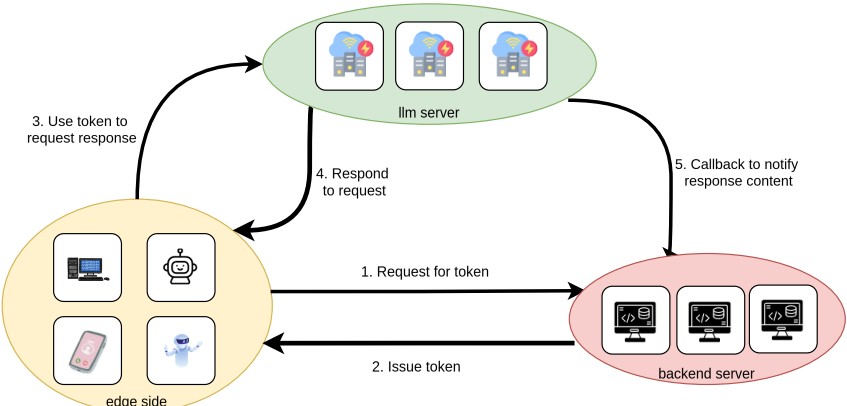

Figure 3: Dynaseal token

4. **Respond to request**: Large model provider responds to edge-side.

5. **Callback to notify response content**: Upon response completion, callback notifies response content.

## 3.4 ATTACK PREVENTION

Our system implements comprehensive security measures to prevent potential attacks:

- **Token Tampering**: Malicious actors may attempt to modify token contents to gain unauthorized access. We prevent this through robust **digital signatures** that ensure token integrity, making any unauthorized modifications detectable.

- **Token Replay**: Attackers might try to reuse previously issued tokens. Our system mitigates this risk by implementing **extremely short validity periods**, rendering captured tokens unusable after expiration.

- **Invalid Model Invocation**: To prevent unauthorized model access or parameter manipulation, tokens contain **critical execution parameters**. The LLM service provider enforces strict invocation constraints based on these embedded parameters, ensuring all calls comply with specified limitations.

## 4 EXPERIMENTS

## 4.1 TRAFFIC CONSUMPTION COMPARISON

We compared the network traffic consumption between LLM service providers and backend servers across different approaches including pre-embedded API keys and server relay, as shown in Table 1. Detailed a test prompt is provided in Appendix A.2.

Table 1: Traffic Flow and Key Deployment Comparison (byte)

| Method | LLM Provider | | Backend Server | | Client | |
|---|---|---|---|---|---|---|
| | **In** | **Out** | **In** | **Out** | **In** | **Out** |
| Pre-embedded API Key | 3411 | 711692 | N/A | N/A | 711692 | 3411 |
| Server Relay | 3411 | 711692 | 715103 | 715103 | 711692 | 3411 |
| Dynaseal | 3503 | 721239 | 887 | 10254 | 712487 | 4118 |

Analysis shows that we have identified the trade-off in existing solutions, which reduces the backend server's traffic consumption by 99% while maintaining the large model service provider's traffic level unchanged, and simultaneously ensuring key security.

## 5 CONCLUSION

We propose a novel method, Dynaseal, allowing backend constraints on model invocation, effectively addressing existing edge-side model invocation security issues while avoiding server relay waste. We provide a complete design and interaction flow, demonstrating the feasibility of this approach.

ACKNOWLEDGMENTS

- Shenzhen Smartlink Technology Co., Ltd sponsored the server and large model resources required for our research.
- This work was supported by Undergraduate Training Programs for Innovation and Entrepreneurship of Shaanxi Province No. S202411664179.
- The author, Jiahao Zhao, participated in the Tencent's T-Spark Program, learning about text to images and large language model, which greatly aided our research.

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

## A  APPENDIX

### A.1  COMPARISON OF DIFFERENT MODEL INVOCATION METHODS

Table 2: Comparison of Different Model Invocation Methods

| API Provider | Client-side key control | Anti-tampering | Critical parameter control |
|---|---|---|---|
| Openai API | No | No | No |
| Zhipu API | Yes | Yes | No |
| OneAPI | No | No | No |
| **Dynaseal(Ours)** | Yes | Yes | Yes |

### A.2  TEST PROMPT CASE

∗∗ Interdisciplinary  Knowledge Integration and Future  Society  Deduction Research Framework∗∗

As Chief Analyst of the Future  Research  Institute , please compile a  strategic  outlook  report  for 2150 integrating  the  following dimensions:

1. ∗∗Fundamental Scientific  Breakthroughs∗∗
− Quantum biology's latest   theoretical  framework (including  mathematical  models)
− Antimatter energy   industrialization   pathway (with  technical  roadmap)
− Three−stage  application of spacetime  structure  engineering in   interstellar   travel

2. **Social Structure Evolution**
- Legal system reconstruction under widespread brain-computer interfaces
- Ethical debates on consciousness uploading (citing >5 philosophical schools)
- Feasibility analysis of cross-species parliamentary systems (including insect civilization communication cases)

3. ** Civilization Form Predictions **
- Galactic civilization taxonomy (extended Kardashev Scale version)
- Resource allocation game theory model in Dyson Swarm construction
- Potential forms and defense strategies of dimensional folding warfare

4. **Techno-Ethical Matrix**
- Regulatory sandbox design for memory editing technology
- Nine-grid model for AI personality rights recognition standards
- Cultural contamination assessment index system for interstellar colonization

**Format Requirements:**
Academic paper structure (Abstract-Literature Review-Methodology-Body-Conclusion)
Each section must contain:
- Core thesis (**bold**)
- Argumentation flowchart
- Counterarguments (red text)
- Case evidence (2103 Mars Federation Case)
Include >20 fictional citations ([Author] "Title" *Journal* Year)
Interactive node every 10k words (e.g.: "Scan hologram here for decision simulation")

**Style Guidelines:**
- Balance rigor and imagination
- Etymological annotations for key terms
- Dual optimistic/pessimistic tech development paths
- Emoji summarizer per paragraph end

**Special Constraints :**
Ban vague terms like "revolutionary"/"disruptive"
Use quantum-state possibility superposition narration
All predictions require self-falsification mechanisms

First present conceptual map (mindmap format), then detailed analysis, concluding with 5D radar chart for risk assessment. Append 10k-word Socratic dialogue examining methodological limitations.

