# OpenReview forum: "Dynaseal: A Backend-Controlled LLM API Key Distribution Scheme with Constrained Invocation Parameters"
_ICLR.cc/2025/Workshop/BuildingTrust — BuildingTrust_

### Official Review · Reviewer_tcwW · 2025-02-21
**Token-Based LLM Invocation for Edge Devices: Promising Results, but Methodology Needs Clarification**

**Rating:** 6
**Confidence:** 2

**Review:**

This paper proposes a novel model invocation approach for edge devices that balances the low latency of direct client-to-LLM-server connections with the security of a relay-based system. Instead of directly authenticating with the LLM server or solely relying on a relay, the proposed approach uses a hybrid method. The client first obtains an authentication token from a relay server (backend). This token is then used by the client to directly query the LLM service. This approach introduces challenges, such as the potential for token reuse which the authors address by implementing techniques like extremely short token validity periods.

To evaluate the effectiveness of their approach, the authors compare it against both direct authentication and traditional relay setups. Their experimental results demonstrate that the proposed method reduces the load on the relay server by 99% compared to a standard relay system while maintaining security comparable to a relay system. The main claim is that by introducing this token-based approach, the authors reduce the load, increase the security and keep the latency low.

Strengths:
- Demonstrates a new model invocation approach where edge devices retrieve authentication tokens from the backend server which is then used to query an LLM service
- Technique demonstrates a substantial reduction in bandwidth usage of the backend server while ensuring that the system is resistant to attacks in ideal situations

Weaknesses:
- It is unclear to me whether the paper is a good fit for this workshop. While it is written with LLMs in mind, the same setup can be used for any generic scenario where a client needs to rely on a separate entity to process a large amount of information.
- In the same vein, it would be ideal if the authors could examine approaches seen in traditional networks in the field of computer science instead of just LLM providers.
- A lot of information is missing on how the experiments were conducted:
  - The authors mention that various LLM service providers were used but do not go into detail about what these are (lines 196). Are these service providers on the web like OpenAI or were these local service providers that were used for the experiment?
  - What was the latency between the various parts of the network like the client, backend and LLM provider?
  - How are these digital signatures created? A simple citation would be sufficient to get an idea.
  - How short are these "extremely short validity periods" for the duration of the authentication tokens and how are they determined? Do they account for latency based on large-scale geographical distance?
- (minor) It's not clear what the values in Table 1 are supposed to represent. I'm assuming the B in the caption refers to bytes but this should be named more explicitly
- (minor) While the 99% reduction in bandwidth usage for the backend is great, it is unclear if this leads to substantial improvements in practice. It would be ideal if the authors could provide citations of such systems where the backend server is the bottleneck.

Question:
- Really minor, but why is the technique called dynaseal?

---

### Official Review · Reviewer_RT6C · 2025-02-24
**Review of "Dynaseal: A Backend-Controlled LLM API Key Distribution Scheme"**

**Rating:** 6
**Confidence:** 3

**Review:**

### Strengths
This paper introduces Dynaseal, a backend-controlled authentication mechanism that replaces static API keys and server relays with cryptographically signed, short-lived JWT tokens. It improves security, efficiency, and scalability by enforcing fine-grained invocation constraints while reducing backend traffic by 99%. The approach is well-motivated, tackling real-world security vulnerabilities in LLM API access. The writing is clear and well-structured, with detailed explanations of token architecture, authentication flow, and attack prevention.

### Weaknesses
The paper lacks rigorous security analysis, with limited discussion of adversarial attacks beyond token replay. While it claims zero additional latency, no empirical latency measurements are provided. The scalability impact of signing/verifying JWT tokens at high loads is unexplored, and the evaluation focuses only on traffic reduction, missing real-world adversarial testing. Clarifications on experiment methodology and comparisons with industry-standard authentication mechanisms (e.g., OAuth 2.0) would strengthen the work.

---

### Official Review · Reviewer_t2qu · 2025-03-02
**Secure Backend-Controlled Access for LLM APIs with Parameterized Invocation Constraints**

**Rating:** 6
**Confidence:** 2

**Review:**

The paper introduces Dynaseal, a framework designed to address security and efficiency challenges in edge-device interactions with cloud-based LLMs. By combining cryptographically signed short-lived JWT tokens with backend-enforced invocation parameters, the authors propose a novel alternative to traditional static API keys and server relays. The core innovation lies in decentralizing authentication while maintaining strict control over model invocation parameters such as model selection, token limits, and expiration times.

The strengths of this work are evident in its practical relevance and security-first design. Dynaseal eliminates the need for resource-heavy server relays, reducing backend traffic by 99% (as shown in Table 1) without introducing latency for edge devices. The integration of JWT tokens with embedded constraints ensures tamper-proof authentication, and the short expiration periods (e.g., 1s) mitigate replay attacks effectively. The security analysis comprehensively addresses attack vectors, including token tampering and unauthorized parameter manipulation, demonstrating a robust defense mechanism. The paper’s structure is logical, with clear visualizations (Figures 1–3) that aid in understanding the protocol’s workflow and token architecture.

However, the work has notable limitations. Performance metrics are narrowly focused on traffic reduction, omitting critical evaluations such as computational overhead for token generation/validation and latency under varying network conditions. The claim of "zero additional latency" lacks empirical validation, particularly in scenarios with unstable connectivity. While short token expiration times enhance security, they risk high timeout rates in real-world environments, yet the paper does not propose adaptive strategies (e.g., dynamic TTL adjustments) to address this trade-off. Furthermore, the comparison with existing methods (Table 2) oversimplifies the landscape; a deeper engagement with related work, such as OAuth 2.0 extensions or federated token systems, would strengthen the novelty claim. Reproducibility concerns also arise due to sparse experimental details (e.g., LLM provider configurations, test prompts), and the hypothetical test case in Appendix A.2 limits confidence in real-world applicability.

The significance of Dynaseal lies in its potential to reshape edge-AI security paradigms. By enabling backend-enforced constraints on LLM invocations, the framework offers a scalable solution for modern deployments. However, broader adoption hinges on addressing network reliability challenges and expanding performance evaluations. Future work should explore adaptive token management, benchmark against state-of-the-art authentication frameworks, and provide open-source implementations to facilitate reproducibility.

---

### Decision · Program_Chairs · 2025-03-04

**Decision:**

Accept

**Comment:**

The reviewers find Dynaseal to be a novel and relevant contribution but also highlight critical gaps in performance evaluation, security analysis, and experimental rigor. The paper is marginally above the acceptance threshold due to its strong motivation and potential impact, but its claims require stronger empirical support especially by benchmarking against industry-standard authentication frameworks.